# Pathophysiological Aspects of Alcohol Metabolism in the Liver

**DOI:** 10.3390/ijms22115717

**Published:** 2021-05-27

**Authors:** Jeongeun Hyun, Jinsol Han, Chanbin Lee, Myunghee Yoon, Youngmi Jung

**Affiliations:** 1Institute of Tissue Regeneration Engineering (ITREN), Dankook University, Cheonan 31116, Korea; j.hyun@dankook.ac.kr; 2Department of Nanobiomedical Science and BK21 PLUS NBM Global Research Center for Regenerative Medicine, Dankook University, Cheonan 31116, Korea; 3Department of Regenerative Dental Medicine, College of Dentistry, Dankook University, Cheonan 31116, Korea; 4Department of Integrated Biological Science, Pusan National University, Pusan 46241, Korea; wlsthf1408@pusan.ac.kr (J.H.); lcb102@pusan.ac.kr (C.L.); 5Department of Surgery, Division of Hepatobiliary and Pancreas Surgery, Biomedical Research Institute, Pusan National University, Pusan 46241, Korea; ymh@pusan.ac.kr; 6Department of Biological Sciences, Pusan National University, Pusan 46241, Korea

**Keywords:** alcoholic liver disease, alcohol metabolism, lipotoxicity, inflammation, fibrosis

## Abstract

Alcoholic liver disease (ALD) is a globally prevalent chronic liver disease caused by chronic or binge consumption of alcohol. The liver is the major organ that metabolizes alcohol; therefore, it is particularly sensitive to alcohol intake. Metabolites and byproducts generated during alcohol metabolism cause liver damage, leading to ALD via several mechanisms, such as impairing lipid metabolism, intensifying inflammatory reactions, and inducing fibrosis. Despite the severity of ALD, the development of novel treatments has been hampered by the lack of animal models that fully mimic human ALD. To overcome the current limitations of ALD studies and therapy development, it is necessary to understand the molecular mechanisms underlying alcohol-induced liver injury. Hence, to provide insights into the progression of ALD, this review examines previous studies conducted on alcohol metabolism in the liver. There is a particular focus on the occurrence of ALD caused by hepatotoxicity originating from alcohol metabolism.

## 1. Introduction

Alcohol has long been closely related to human culture, used as a drug, and considered an indulgence [1,2]. As the consumption of alcoholic beverages increases, it contributes to the significant elevation of morbidity and mortality worldwide [3,4]. In 2018, the World Health Organization (WHO) estimated that alcohol consumption is responsible for three million deaths worldwide annually, accounting for nearly 14% of the total mortality in people aged 20 to 40 years old [5]. In addition to social and psychiatric problems, more than 200 diseases that affect several organs, such as the brain, heart, gastrointestinal tract, and liver, are related to habitual alcohol consumption [6,7,8]. Among these alcohol-damaged organs, the liver is specifically susceptible to damage because the liver is the primary site of alcohol metabolism in the body [9,10]. Alcohol metabolism generates products that damage the liver, resulting in alcoholic liver disease (ALD), a main cause of chronic liver disease [4,9,10,11]. ALD encompasses a broad spectrum of conditions, including alcoholic fatty liver (simple steatosis), alcoholic hepatitis, alcoholic cirrhosis, and liver cancer [4,11,12,13]. Alcoholic fatty liver is defined by fat accumulation in hepatocytes without substantial inflammation, or hepatic fibrosis, and is observed in up to 90% of heavy drinkers [9,11,12,13,14]. Heavy long-term alcohol consumption accelerates the transition of alcoholic fatty liver into alcoholic hepatitis, which is characterized by steatosis, ballooning hepatocytes, and infiltration of neutrophils with or without fibrosis [11,12,13,14,15,16]. Alcoholic cirrhosis is a more severe form of ALD and is defined by disorganized liver architecture with fibrosis [11,12,13,14,15,16].

Although public awareness of the hepatotoxicity caused by alcohol is rising, the trend of alcohol consumption is steadily increasing [4,17]. Alcoholic cirrhosis accounts for approximately half of the deaths related to liver cirrhosis, and alcohol is known to accelerate liver injury in people infected with hepatitis virus [18,19]. To prevent and/or reverse ALD progression, abstinence from alcohol is the most fundamental solution, but it is difficult to suppress the recurrence of drinking because most patients with ALD are alcoholics [20]. Liver transplantation is considered the definitive treatment for ALD, like other end-stage liver diseases [21,22]. However, liver transplantation is a complicated treatment option because it depends on donor availability and demands abstinence from alcohol for at least six months before and after the transplant [23]. There is also an ethical concern that the transplanted liver could be “wasted” on a patient who eventually relapses to drinking and damages the transplanted liver [21]. In addition, there is no effective Food and Drug Administration (FDA)-approved drug for treating patients with ALD [24]. Therefore, it is important to develop effective therapeutics for ALD. Understanding the process of alcohol metabolism in the liver and the actions of hepatotoxic intermediates involved in ALD progression could provide clues for developing ALD treatment strategies. Herein, the metabolic processes of alcohol are summarized, and ALD pathogenesis is reviewed, providing knowledge into the underlying mechanism of ALD.

## 2. Alcohol Metabolism

Ingested alcohol is absorbed through the stomach and intestines. Less than 10% of absorbed alcohol is excreted in breath, sweat, and urine. This means that more than 90% of the absorbed alcohol circulates through the body and is eventually transported to the liver via the portal vein [10,25]. Due to the high levels of alcohol metabolizing enzymes in the liver, the liver plays a major role in alcohol metabolism [10]. In the liver, alcohol is metabolized by oxidative and non-oxidative pathways (Figure 1) [26,27]. The oxidative pathway is the major pathway for alcohol metabolism and is composed of two steps. First, alcohol is oxidized to acetaldehyde by alcohol dehydrogenase (ADH), a primary enzyme that converts alcohol to acetaldehyde [28]. Excess alcohol consumption increases the expression and activity of cytochrome P450 2E1 (CYP2E1), not ADH. Activated CYP2E1 promotes the production of acetaldehyde through the formation of reactive oxygen species (ROS) [29,30]. In addition, peroxisomal catalase breaks down alcohol to acetaldehyde, but its action is considered a minor pathway because of its small contribution to alcohol digestion [10,27]. The second step in the oxidative pathway involves the rapid conversion of acetaldehyde to acetate by aldehyde dehydrogenase (ALDH). Acetate is metabolized into carbon dioxide (CO_2_), fatty acids (FAs), and water (H_2_O) in peripheral tissues, not the liver [10,27]. The nonoxidative pathway accounts for a minor portion of alcohol metabolism in quantitative terms [31,32,33]. A small amount of alcohol is nonoxidatively conjugated to various endogenous metabolites by different enzymes. For example, enzymatic esterification of alcohol with FAs forms fatty acid ethyl ester (FAEE) and phospholipase D (PLD) catalyzes transphosphatidylation of phosphatidylcholine with ethanol to form phosphatidylethanol (PEth). In addition, alcohol conjugated to glucuronic acid and sulfate generates ethyl glucuronide (EtG) and ethyl sulfate (EtS), respectively [31,32,33].

Products generated during alcohol metabolism damage the liver and act as a driving force of ALD progression from alcoholic steatosis to alcoholic cirrhosis [34,35]. The most well-known toxic compound produced by alcohol metabolism is acetaldehyde [36]. Acetaldehyde interacts directly with DNA and causes point mutations and chromosomal damage. It also binds to a variety of proteins to form acetaldehyde adducts, which distorts liver function and structure [37,38]. These protein adducts upregulate CYP2E1 expression and enhance oxidative stress [39,40]. In addition, Holstege et al. [41] demonstrated that protein adducts contribute to lipid accumulation, inflammation, and fibrosis, playing a key role in the pathogenesis of various stages of ALD. Acetate has been reported to increase portal blood flow by circulating in the bloodstream, although it is less toxic than acetaldehyde [42]. Nonoxidative pathway-derived metabolites, including PEth and FAEE, are also known to cause alcohol toxicity, but the mechanisms have not yet been determined [31,43].

## 3. Oxidative Ethanol Metabolites Are Involved in ALD Pathogenesis

Oxidative ethanol-derived metabolites exert a broad spectrum of damage in the liver, ranging from lipid accumulation in hepatocytes to inflammation, fibrosis, and carcinogenesis (Figure 2). Excessive lipid accumulation in hepatocytes results in massive hepatocyte death, which triggers pro-inflammatory and pro-fibrogenic responses, increasing the risk of liver cancer [44]. Because there is a dearth of information about ALD pathogenesis exerted by non-oxidative ethanol metabolism, this section reviews the mechanisms underlying oxidative ethanol metabolite-mediated hepatotoxicity caused by excess alcohol exposure [45].

### 3.1. Excessive Lipid Accumulation in Hepatocytes

Hepatocytes accumulate lipids when there is an oversupply of lipids and the lipid removal pathway is impaired [46]. The former occurs when the liver increases uptake of exogenous FAs or lipid production in the liver, called de novo lipogenesis. Acetaldehyde, in particular, promotes lipolysis in adipose tissue and elevates the amount of free FAs (FFAs) that are absorbed by the liver [15,47]. The hepatic FA transporters, such as FA transporter proteins (FATPs) and FA translocase (FAT/CD36), are involved in this process, and upregulation of FATP2, FATP5, and FAT/CD36 has been observed in liver tissues in rodent models of ALD [48,49]. In addition, acetaldehyde dysregulates the 5′ adenosine monophosphate-activated protein kinase (AMPK) pathway, which regulates the expression of lipogenic transcription factors, including sterol regulatory element-binding protein 1c (SREBP-1c) and carbohydrate-responsive element-binding protein (ChREBP) [50]. Thus, there is an increase in the expression of lipogenic enzyme genes, such as acetyl-CoA carboxylase 1 (ACC1), fatty acid synthase (FASN), and sterol-CoA desaturase 1 (SCD1) [15,51]. LIPIN1 plays a role in lipid homeostasis. Hepatic LIPIN1 either acts as a transcriptional coactivator (LIPIN1α) in the nucleus or functions as a Mg^2+^-dependent phosphatidate phosphatase (LIPIN1β) promoting biosynthesis of triglyceride and phospholipid in the cytoplasm [52]. It has been reported that ethanol upregulates LIPIN1 via the activation of SREBP-1c in an AMPK-dependent manner, leading to increased lipid accumulation in hepatocytes [52].

Lipid clearance in the liver is mediated by mitochondrial β-oxidation and secretion of excessive stored triglycerides that are packed into very-low density lipoprotein (VLDL) [53]. It should be noted that alcohol-induced dysregulation of mitochondrial β-oxidation is considered the most significant contribution to lipid accumulation in the liver [46]. Splicing factor arginine/serine-rich 10 (SFRS10), an RNA splicing factor, promotes the skipping of the LIPIN1 exon and generates the LIPIN1α variant [52]. Yin et al. [54,55] demonstrated that sirtuin 1 (SIRT1) activating SFRS 10 is inhibited by ethanol via upregulation of microRNA-217, and that suppression of SIRT1 increases the LIPIN1β/α ratio, enhancing the lipid biosynthetic activity of LIPIN1β and reducing its capacity for mitochondrial FA β-oxidation. In addition, overexpression of LIPIN1 in mice has been shown to alleviate VLDL secretion without altering two essential proteins for VLDL formation: apolipoprotein B (ApoB) and microsomal triglyceride transfer protein (MTTP) [52]. Peroxisome proliferator-activated receptor alpha (PPARα), a transcriptional factor, regulates the expression of genes participating in FA oxidation in the mitochondria [56]. Acetaldehyde inactivates PPARα by suppressing the DNA binding affinity of PPARα [57]. Additionally, inactivated PPARα inhibits transcription of FA oxidation-related genes, including the gene encoding carnitine palmitoyltransferase 1 (CPT1), a rate-limiting enzyme of mitochondrial β-oxidation, by translocating FA into mitochondria [58]. Autophagy is also an important mechanism for removing lipid droplets accumulated by ethanol. Acute exposure to ethanol activates autophagy by inhibiting mammalian targets of rapamycin (mTOR) [59]. However, chronic exposure to ethanol inhibits autophagy, which seems to be due to a decrease in AMPK activity caused by acetaldehyde [60].

The ethanol-derived toxic metabolites dysregulate multiple aspects of hepatic lipid metabolism—they increase hepatic FA uptake and de novo lipid synthesis and decrease FA oxidation and lipid export. The effects converge to cause hepatocellular lipid accumulation. Since hepatocellular lipid accumulation is the earliest sign of ALD, further study on lipid metabolism in hepatocytes could create opportunities for early therapeutic intervention for people at risk of advanced ALD.

### 3.2. Dysregulated Immune System

Accumulating evidence indicates that chronic alcohol consumption disrupts the intestinal barrier and the tight and adherent junctions in the colonic mucosa, which promotes the translocation of lipopolysaccharide (LPS) to the circulation [61,62,63]. The detrimental effects of acetaldehyde contribute to this via upregulation of microRNA-212 in enterocytes and downregulation of zonula occludens 1 (ZO-1), a tight junction component [64]. Both acetaldehyde and LPS activate Kupffer cells, the liver-resident macrophages, to release ROS and chemokines that recruit bone marrow-derived neutrophils and blood-derived monocytes into the liver [15,65]. This stage of ALD is called alcoholic hepatitis. ROS released from Kupffer cells mediates the activation of the toll-like receptor 4 (TLR4)/mitogen-activated protein kinase (MAPK)/nuclear factor (NF)-κB axis [66]. It has also been reported that FFA promotes the activation of the TLR4-mediated NF-κB signaling pathway and inflammatory cyclooxygenase 2 (COX2) expression in macrophages in vitro [67]. Furthermore, it has been demonstrated in in vivo experiments; TLR4 knockout (KO) mice were found to be resistant to alcohol-induced hepatic steatosis [68,69].

The levels of pro-inflammatory cytokines, such as tumor necrosis factor alpha (TNFα), interleukin (IL)-1β, and IL-8, are upregulated in patients with ALD [70]. When exposed to oxidative ethanol metabolites, such as acetaldehyde and acetate, the NF-κB signaling pathway is activated and the level of TNFα is elevated in rodent macrophages [71]. In a mouse model of ALD generated by chronic alcohol consumption, the NF-κB pathway was activated in Kupffer cells, and systemic release of pro-inflammatory factors, including TNFα, IL-6, and macrophage chemoattractant protein 1 (MCP1), was observed [71]. Inhibition of SIRT1, an NF-κB antagonist, also partly induces the activation of NF-κB signaling. The NF-κB signaling mediates ROS-triggered inflammatory responses via downstream effectors [72], such as intercellular adhesion molecule 1 (ICAM1). ICAM1 is involved in the contact of hepatocytes with neutrophils, promoting neutrophil-mediated hepatocyte killing (Figure 3) [73].

Oxidative ethanol-derived metabolites may influence adaptive immunity and impair antigen presentation, which is required for T- and B-cell activation [74]. The amounts of circulating antibodies against hydroxyethyl radicals (HER)-protein adducts and lipid peroxidation-derived aldehydes, such as malondialdehyde (MDA), are elevated in patients with alcoholic hepatitis [74]. The increased levels of anti-HER and anti-MDA antibodies are associated with the presence of activated CD4+ T cells in peripheral blood. The macrophage’s proteasome function is also altered by oxidative ethanol metabolism, which consequently reduces antigen presentation [60,75]. Ethanol inhibits antigen presentation in macrophages and dendritic cells [76]. Chronic alcohol ingestion decreases the number of F4/80+ cells expressing major histocompatibility complex (MHC)-I and MHC-II [77].

FAs sensitize non-immune cells, such as hepatocytes, to pro-inflammatory signals and impair their ability to respond to hepatoprotective signal proteins, such as signal transducer and activator of transcription 3 (STAT3) [78,79]. Furthermore, the production of ROS during ethanol metabolism rapidly increases the fluidity of the hepatocyte cell membrane and promotes cytoplasmic iron overload [80] and accelerated lipid peroxidation, eventually leading to massive hepatocyte death.

### 3.3. Induction of Fibrosis

Massive hepatocyte death provokes the fibrotic repair process in the liver [81]. When the hepatic parenchyma is progressively replaced by scar tissue, the metabolic function of the liver is compromised, leading to liver failure [15]. Hepatic stellate cells (HSCs) are activated in the livers of patients with alcoholic steatohepatitis to become the main producers of extracellular matrix proteins, such as collagens and fibronectin, causing liver fibrosis [15]. The fibrogenic mechanisms are initiated and perpetuated by alcohol metabolism (Figure 3) [11]. The protein adducts generated by ethanol-derived acetaldehyde and the aldehydes produced from lipid peroxidation, such as MDA, turn on pro-fibrogenic pathways in activated HSCs [11,15]. It has been shown that the ALDH2-KO mouse having highly accumulated acetaldehyde in the liver is more susceptible to alcohol-induced liver inflammation and fibrosis compared to wide-type mice because of the higher level of MDA-acetaldehyde adducts [82]. The results of an in vitro experiment showed that acetaldehyde released from hepatocytes entered HSCs and directly led HSCs to express type I collagen genes [83]. Acetaldehyde regulates the expression of collagen genes in a protein kinase C (PKC)-dependent manner [84]. In human HSCs, PKC phosphorylates p70 S6 kinase (p70S6K) by activating extracellular signaling-regulated kinase (ERK) and phosphoinositide 3 kinase (PI3K), leading to collagen gene expression [84]. Another pro-fibrogenic mechanism stimulated by acetaldehyde is the transforming growth factor beta (TGFβ) pathway, which is essential in liver fibrogenesis [85]. In human HSCs treated with acetaldehyde, the transcription levels of type I collagen and fibronectin genes were found to be elevated in a TGFβ-independent manner in the early phase of treatment [86,87], while TGFβ-dependent responses, including the secretion of latent TGFβ1 and the expression of type II TGFβ receptor, occurred in the late phase [85]. LPS from the gastrointestinal tract also contributes to HSC activation by increasing the susceptibility of HSCs to acetaldehyde and TGFβ [88,89,90]. Moreover, acetaldehyde has been shown to inhibit β-catenin phosphorylation and degradation by blocking glycogen synthase kinase (GSK)3B, thereby the active β-catenin translocalizes into the nucleus and upregulates the expression of fibrogenic genes [91].

An accumulating body of evidence indicates that acetaldehyde activates HSCs in ALD by inducing oxidative stress [60]. ROS generated during CYP2E1-dependent oxidative ethanol metabolism enhance collagen production in HSCs co-cultured with hepatocytes [92]. Hydrogen peroxide (H_2_O_2_), an ROS, also aids acetaldehyde-mediated hepatic fibrogenesis. DNA binding affinity and activity of CCAT/enhanced binding protein beta (C/EBPβ) that promotes the transcription of collagen I α1 (COLI α1) are increased by acetaldehyde, and the actions of C/EBPβ are mediated by H_2_O_2_ [93]. Similarly, transcriptional activity of peroxisome proliferator-activated receptor gamma (PPARγ) is inhibited by acetaldehyde in the H_2_O_2_-dependent pathway in activated HSCs [94]. In addition, acetaldehyde binds to glutathione (GSH) and weakens its antioxidant activity [95]. Nuclear erythroid 2-related factor 2 (NRF2), a transcription factor activated by oxidative stress, is presented to upregulate antioxidant gene expression [96]. Thus, Nrf2 overexpression in mice improves ALD by preventing oxidative stress [97], whereas Nrf2 deficiency promotes ALD progression [98].

Moreover, chronic alcohol consumption suppresses the anti-fibrotic function of natural killer cells that are cytotoxic to activated HSCs, accelerating hepatic fibrogenesis. This suppression is mediated by TNF-related apoptosis-inducing ligand (TRAIL)-TRAIL receptor interaction and interferon gamma (IFNγ) [78,99]. In addition, IL-22 released by natural killer cells and T helper cells is shown to inhibit activation and proliferation of HSCs stimulated by acetaldehyde [100]. It was demonstrated that IL-22 promoted the nuclear translocation of Nrf2 in HSCs and led to HSC inactivation by arresting them at G1/S phase. Overexpression of IL-22 was shown to induce HSC senescence in carbon tetrachloride (CCl_4_)-induced fibrotic livers of mice by upregulating p53, a mediator of cellular senescence [101]. IL-22 also has anti-apoptotic, anti-oxidative, and pro-regenerative effects against alcohol-induced liver damage, and drives the onset of clinical trials of IL-22 for the treatment of ALD [102].

The ROS also stimulate intracellular pro-fibrogenic pathways in HSCs, including the ERK, protein kinase B (PKB/Akt), c-Jun N-terminal kinase (JNK), and tissue inhibitor of metalloproteinase 1 (TIMP1) pathways. Whereas, the activity of matrix metalloproteinases (MMPs) is inhibited by ROS [60]. The pro-fibrotic effect of ROS was confirmed by the finding that ROS-scavenging enzymes prevent hepatic fibrosis in animal models of ALD [103]. In addition, ethanol-mediated lipid peroxidation is blocked and liver injury is reduced in *Cyp2e1*-KO mice [104]. These findings suggest that the suppression of oxidative stress, potentially by antioxidants, could attenuate ALD-related fibrosis. For example, helenalin isolated from *Centipeda minima* prevented liver injury and fibrosis in rats with intragastric alcohol infusions up to 24 weeks by enhancing the activities of ethanol-detoxifying enzymes, ADH and ALDH [105]. Helenalin also reduced CYP2E1 activity, inhibited HSC activation by suppressing the TGFβ signaling pathway, and decreased fibrosis by upregulating the levels of MMP1 and MMP9. Similarly, other antioxidants, such as hesperidin and fraxetin, were shown to attenuate alcohol-induced HSC activation and fibrosis [106,107]. However, any pharmacological treatment has not been successful in clinical trials of patients with alcoholic cirrhosis [108].

### 3.4. Development of Cancer

Alcohol is recognized as a carcinogen by the International Agency for Research on Cancer. This is because alcohol causes many types of cancers, including hepatocellular carcinoma (HCC) in humans [109]. It is known that excessive alcohol consumption leads to a 3- to 10-fold increase in the risk of HCC and accounts for 30% of HCC cases worldwide [109,110]. Several mechanisms by which alcohol induces carcinogenesis in the liver have been suggested. Among them, the carcinogenic effect of ethanol-derived acetaldehyde forming protein and DNA adducts may be specific to ALD-associated HCC [111,112]. For example, N2-ethyl-deoxyguanosine (N2-Et-dG), an acetaldehyde-DNA adduct, has been detected in patients with ALD [113]. It has also been reported that N2-propano-2′-deoxyguanosine (N2-Et-dGTP), another DNA adduct, changes DNA integrity [114]. Moreover, acetaldehyde directly binds to amino acids, and the resulting protein adduct (with O6-methylguanine methyltransferase) interferes with the DNA repair system, contributing to liver carcinogenesis [115]. It has been shown in animal studies that acetaldehyde increases point mutation frequency in the hypoxanthine phosphoribosyl transferase (HPRT) gene in lymphocytes, a biomarker of lymphocyte functional deficiency [116].

Genetic variations in alcohol-metabolizing enzymes are the main causes of carcinogenic acetaldehyde production, and thus have potential as biomarkers of alcohol-induced liver cancer [117]. In particular, a deficiency of mitochondrial ALDH2 activity leads to extreme accumulation of acetaldehyde because ALDH2 is a key acetaldehyde-detoxifying enzyme. The ALDH2 mutant genotype, *ALDH2*2* allele (also known as *ALDH2(E487K)* or *ALDH2 rs671* polymorphism), has been found to be associated with a high incidence of liver cancer in heavy drinkers [118,119,120]. The relationship between ALDH2 deficiency and the development of liver cancer was recently proven. Seo et al. [121] showed that after exposure to carbon tetrachloride (a hepatotoxin) and ethanol, *Aldh2*-deficient hepatocytes produced abundant oxidized mitochondrial DNA that was favorable for HCC progression by activation of oncogenic pathways together with acetaldehyde. They also demonstrated that ALDH2 deficiency increases the risk of HCC development in patients who have hepatitis B virus-cirrhosis and abuse alcohol. These findings might explain why the prevalence of ALD-associated liver cancer is high in the Asian population, among whom 30–40% have an ALDH2 deficiency. In addition, the homozygous *ADH1C*1* allele has been discovered to be a predictor of an increased risk of HCC in ALD patients, since the *ADH1C*1* allele encodes a 2.5-fold higher ethanol oxidizing capacity to generate more acetaldehyde than the *ADH1C*2* allele [122].

Besides the accumulation of harmful acetaldehyde, alcohol can induce hepatocarcinogenesis by promoting oxidative stress contingent on ROS production from alcohol metabolism [109,110]. Accumulated ROS generate lipid peroxides, such as 4-hydroxy-nonenal (4-HNE), and alter gene expression, leading to the upregulation of pro-inflammatory cytokines and the activation of immune cells [109]. ROS also upregulate angiogenesis and the metastatic process [109,123].

## 4. Models Used in the Study of Pathophysiology Related to Alcohol Metabolism

Progress in research and treatment of ALD is slowed, at least in part, due to the lack of animal models that fully reflect the spectrum of human ALD. Although various animal models such as primates and micropigs have been proposed for ALD study, rodent models are commonly used because of better handiness in experiments and efficiency in terms of cost and time [124]. However, there are fundamental differences in physiological and biological processes for alcohol between rodents and human [125]. Unlike humans, rodents rarely increase the amount of alcohol consumption over time and stop drinking when blood acetaldehyde level is upregulated [126]. Additionally, rodents have a natural aversion to alcohol. Notably, it is known that the alcohol-catabolizing rate is much faster in rodents than humans. In addition, neutrophil infiltration is hardly detected in rodents during ALD pathogenesis, whereas it is one of the key features of alcoholic steatohepatitis in humans [73]. Furthermore, the sex, age, and genetic background of the animal, and even the animal facility environment, impact susceptibility to alcohol-induced liver damage, and bring to the high inter-variability in each experimental animal, hindering the generation of rodent models of ALD [127]. Therefore, scientists have been developing rodent models which could minimize the differences between rodent models of ALD and human ALD (Table 1). The simplest model of ALD is to give the rodent alcohol-mixed drinking water with normal chow-diet and let them drink it ad libitum. However, this model brings a slight increase of blood alcohol levels (BALs) and mild liver injury [128,129]. DeCarli and Lieber [130,131] developed the alcohol-containing liquid diet formula, called the Lieber-DeCarli liquid diet. Rats and mice are fed ad libitum the Lieber–DeCarli liquid diet without any other foods or drinks. The Lieber–DeCarli liquid diet partially overcomes the rodent’s aversion to alcohol and induces elevated aminotransferases and hepatic steatosis. However, inflammation is mild and fibrosis occurs rarely in the model [132]. The fundamental limitation of the ad libitum alcohol feeding models is that animals do not drink alcoholic water as well as humans do. Hence, the intragastric infusion model, called the Tsukamoto–French (TF) model, was developed by Tsukamoto et al. [133,134]. In this model, alcohol is directly injected into rodents through a surgically implanted intragastric cannula, and higher BALs and more severe liver injuries than ad libitum alcohol-feeding models are observed. However, the levels of hepatic inflammation and fibrosis in TF model are still less than them in human alcoholic hepatitis. Moreover, requirements of technical competence, intensive medical care, and expensive equipment are major obstacles to reproducing the TF model [135]. In 2013, the Dr. Gao group at the National Institute on Alcohol Abuse and Alcoholism of National Institute of Health [136] reported a mouse model of chronic and binge alcohol feeding, termed the NIAAA model (commonly referred to as the Gao-binge model). The Gao-binge model is produced by the Lieber–DeCarli liquid diet for many consecutive days (>10 days) followed by single or multiple binge(s). According to the periods of ethanol feeding, high BALs, elevated level of serum alanine aminotransferase hepatic neutrophil infiltration, and even severe liver damage can be generated. To evaluate the effects of alcohol on late phase of ALD such as cirrhosis or liver cancer, a combination of the Lieber–DeCarli liquid diet with other hepatotoxins, such as CCl_4_, diethylnitrosamine (DEN) or LPS, as a “second hit”, is given to rodents [137,138,139].

In human ALD, blood acetaldehyde level is significantly upregulated and its increase parallels the elevation of ethanol concentration [140,141,142,143]. The high level of blood acetaldehyde in human alcoholics after ethanol ingestion is caused by the decreased oxidation of acetaldehyde rather than its increased formation from ethanol. It is associated with low hepatic ALDH activity in alcoholic subjects [144]. Although the acetaldehyde level also increases in rodents after ethanol treatment, its amount varies depending on the dose of alcohol and duration of alcohol feeding. ALDH2-KO mice may recapitulate the ALDH2*2 allele that encodes ALDH2 with low acetaldehyde-oxidizing activity. As mentioned above, ALDH2-KO mice have higher levels of acetaldehyde and acetaldehyde adducts in blood and/or liver than wild-type mice when exposed to ethanol [145]. It is reported that gut permeability, the concentration of plasma transaminases, and the amount of hepatic triglyceride are higher in heterozygous ALDH2-KO (ALDH2+/−) mice fed the Lieber–DeCarli liquid diet than in alcohol-fed wild-type mice [146]. In addition, ALDH2-KO mice have been shown to be more sensitive to ethanol-induced liver inflammation, fibrosis, and HCC, except hepatic steatosis, than wild-type mice [82,121]. Genes responsible for alcohol oxidization, such as ADH1 and CYP2E1, are also targeted for modification in mice in order to investigate the role of these enzymes in ALD pathogenesis. The ADH1 global KO mice was shown to have a problem in blood ethanol clearance [147], but the effect of knock-out-ADH1 on ALD has been poorly studied. The genetic deficiency of CYP2E1 rarely induced hepatic steatosis and oxidative stress in mice fed a high-fat Lieber–DeCarli liquid diet, whereas overexpression of CYP2E1 increased serum transaminases and accelerated other liver damages, such as upregulation of fibrosis markers including α-smooth muscle actin (α-SMA) and type I collagen, and elevated necroinflammation [104,148].

Advances in bioengineering techniques lead to the development of in vitro and ex vivo models which possibly compromise the limitations that animal models have. To study ALD pathogenesis, Lee et al. [149] developed a three-dimensional in vitro model of ALD by co-culturing rat primary hepatocytes and HSCs in a microfluidic chip where the cells were exposed to ethanol via the flow. Recently, Wang et al. [150] generated the expandable hepatic organoids derived from human embryonic stem cells. These organoids phenocopied the bipotent liver progenitor cells that could differentiate into both hepatocytes and cholangiocytes. The hepatic organoids co-cultured with human fetal liver mesenchymal cells reflected the pathophysiology of human ALD by showing the increased CYP2E1 activity, expression of fibrosis markers, fibrous accumulation, pro-collagen secretion, oxidative stress, steatosis, and expression of lipogenic genes after ethanol exposure. In addition, secretome analysis for inflammatory mediators showed that a number of pro-inflammatory cytokines were upregulated in this model after alcohol treatment. Therefore, these findings suggest that in vitro or ex vivo pathophysiological models are promising to study ALD. However, further research is needed to verify these models and to standardize the culture conditions.

## 5. Conclusions

Consumption of alcohol is deeply ingrained in the daily lives of many people; therefore, it is difficult to suppress consumption and intervene in the progression of ALD. Nevertheless, efforts to prevent and treat ALD by intervening in alcohol-related diseases are considered urgent tasks because of the increasing incidence of ALD and the absence of FDA-approved therapy. To reverse or inhibit the progression of ALD, identification of ALD risk factors and of mechanisms explaining their actions in ALD progression provides vital information for developing therapeutic targets for ALD. Therefore, in this review, we have briefly explained alcohol metabolism and its products, and reviewed the effects of these metabolites in ALD progression. We focused on the hepatotoxicity of ethanol-derived metabolites produced by oxidative alcohol metabolism (acetaldehyde and ROS) during excessive exposure to alcohol. As discussed above, the products of alcohol metabolism directly or indirectly damage the liver, leading to liver cancer. However, the mechanisms of ALD pathogenesis remain unclear because of the lack of ALD patient samples and reliable animal models for ALD that reflect human ALD. Limited understanding of ALD hampers the development of novel therapies for ALD. Even though possible therapeutic candidates have been proposed in many studies based on animal models, their effects are controversial or not proven in the clinical stage. Therefore, further studies are necessary to elucidate the detailed mechanisms of ALD and to discover potential therapeutic targets for treating ALD.

## Figures and Tables

**Figure 1 ijms-22-05717-f001:**
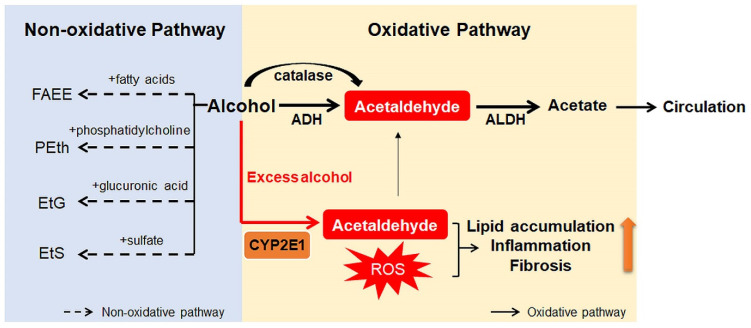
Scheme for alcohol metabolism in the liver. In the liver, alcohol is metabolized by the oxidative and non-oxidative pathway. In the oxidative pathway, the major pathway of alcohol digestion, alcohol is oxidized to acetaldehyde by various enzymes including alcohol dehydrogenase (ADH), cytochrome P450 2E1 (CYP2E1), and catalase. Then, acetaldehyde is broken down into acetate, which is excreted out of the liver. Especially when excessive alcohol is consumed, CYP2E1 is activated and promotes formation of reactive oxygen species (ROS). The non-oxidative pathway accounts for a small amount of alcohol metabolism. Various enzymes nonoxidatively conjugates alcohol with different endogenous metabolites, producing fatty acid ethyl ester (FAEE), phosphatidylethanol (PEth), ethyl glucuronide (EtG), and ethyl sulfate (EtS). The byproducts generated during alcohol metabolism injure the liver by increasing lipid accumulation, inflammation, and fibrosis. Especially, acetaldehyde, the first metabolite of alcohol metabolism, is well known for toxic compounds. ROS, which are generated by activation of CYP2E1, are also considered as one of the major contributors of liver damage. In addition, acetate and non-oxidative metabolites are known to damage the liver.

**Figure 2 ijms-22-05717-f002:**
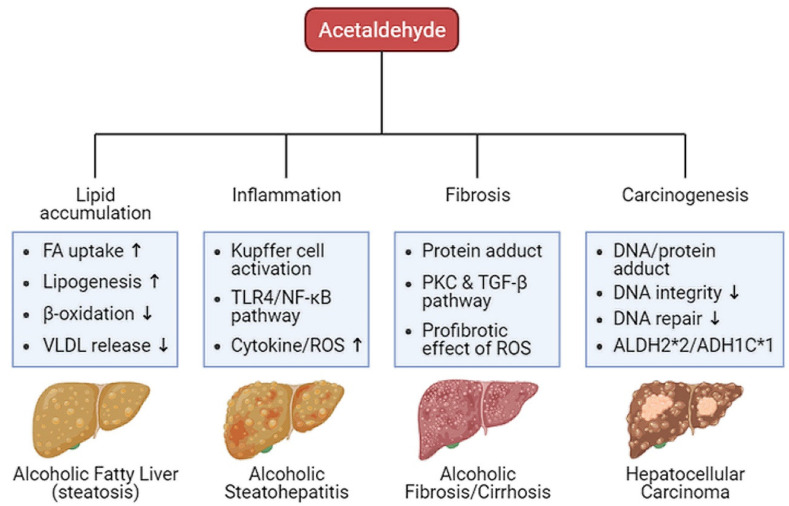
Harmful effects of acetaldehyde on ALD. Acetaldehyde, one of the oxidative ethanol-derived metabolites, exerts a broad spectrum of damage to the liver, ranging from lipid accumulation in hepatocytes to inflammation, fibrosis, and carcinogenesis. The hepatocytes accumulate lipids by the lipid oversupply (increase of fatty acid (FA) uptake to the liver and de novo lipogenesis in the liver) and/or the impaired pathway of lipid clearance (decrease of mitochondrial β-oxidation and secretion of excessive lipids in very-low density lipoprotein (VLDL)). In addition, acetaldehyde activates Kupffer cells, the liver-resident macrophage, to release reactive oxygen species (ROS) and cytokines that recruit other immune cells. When exposed to acetaldehyde, Kupffer cells activate the toll-like receptor 4 (TLR4)-mediated nuclear factor (NF)-κB signaling pathway, triggering inflammatory responses. The acetaldehyde-protein adducts promote the collagen production by activated hepatic stellate cells (HSCs) via the protein kinase C (PKC) and the TGF-β signaling pathway. ROS-mediated oxidative stress also accelerates liver fibrosis. Moreover, protein and DNA adducts with acetaldehyde cause hepatic carcinogenesis. They weaken DNA integrity and interfere with DNA repair system, increasing carcinogenic DNA mutation. Genetic variations in the alcohol-metabolizing enzymes, such as acetaldehyde dehydrogenase (ALDH)2*2 and alcohol dehydrogenase (ADH)1C*1 alleles, suppress the activity of the enzymes, enhancing the amount of acetaldehyde.

**Figure 3 ijms-22-05717-f003:**
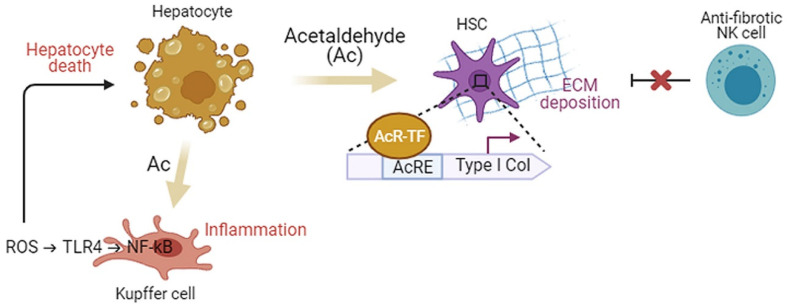
Cellular interactions during the pathogenesis of ALD. Acetaldehyde (Ac) produced by hepatocytes during ethanol oxidation activates Kupffer cells to release reactive oxygen species (ROS) which triggers inflammatory responses in a TLR4/NF-κB-dependent manner and leads to massive hepatocyte death. In vitro, Ac secreted from hepatocytes enters into the hepatic stellate cells (HSCs), promotes the binding of Ac-responsive transcription factors (AcR-TFs) to the acetaldehyde-responsive element (AcRE) in the promoter region of type I collagen (Col) genes to onset of transcription. Ac also activates HSCs via other pro-fibrogenic mechanisms, increasing deposition of extracellular matrix (ECM) proteins. Anti-fibrotic effect of natural killer (NK) cells that are cytotoxic to activated HSCs is inhibited by chronic alcohol consumption.

**Table 1 ijms-22-05717-t001:** Models of alcohol administration to rodent.

Models	Administration	Characteristics	Feasibility
Ad libitum alcohol-drinking water	Oral alcohol consumption by drinking water	Low BAL; minimal elevation of ALT; mild steatosis	Easy to perform
Ad libitum liquid diet (Lieber–DeCarli diet)	Oral alcohol consumption with alcohol-containing liquid diet formula but with no other food or drink	Variable elevation range of ALT; marked steatosis; mild inflammation	Easy to perform
Intragastric infusion (The Tsukamoto-French model)	Direct enteral feeding through a surgically implanted intragastric cannula	High BAL; marked elevation of ALT; severe steatosis; mild inflammation; fibrosis	Difficult to perform
Chronic and binge alcohol feeding (Gao-binge model)	A single or repeated intragastric gavage of alcohol following chronic feeding with the Lieber–DeCarli liquid diet	High BAL; marked elevation of ALT; steatosis; neutrophil infiltration; necrosis; no fibrosis	Easy to perform
Lieber–DeCarli diet + other hepatotoxins (Second hit model)	Addition of hepatotoxins such as DEN, LPS or CCl_4_ during the chronic feeding phase of the Lieber–DeCarli liquid diet	Marked elevation of ALT; high mortality rate; significant liver fibrosis	Easy to perform

ALT, alanine aminotransferase; BAL, blood alcohol level; CCl_4_, carbon tetrachloride; DEN, diethylnitrosamine; LPS, lipopolysaccharide.

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
