# Peer review of "Pathophysiological Aspects of Alcohol Metabolism in the Liver"

_ijms, 2021, doi:10.3390/ijms22115717_

Round 1

Reviewer 1 Report

This is an interesting and readable review of the pathophysiology of alcohol metabolism in the liver.  It provides a good starting place for scientists interested in this area.  The references seem appropriate.  My only significant concern is that the abstract and conclusion both mention the lack of animal models that fully mimic human alcoholic liver disease (ALD).  However, there is little coverage of this in the text and in places there is a lack of distinction between rodent and human data.  A paragraph of section summarizing the existing models and their limitations would be useful.  Is the problem because metabolic pathways differ in rodents and humans?  What other animal models have been studied?  Differences between in vivo and in vitro models might also be useful.  Can normal human hepatocytes be used as a model?  This coverage might lead to a table of models.

Author Response

Thank you for your comment. As your suggestion, we introduced the existing alcoholic liver disease models and their limitations, focusing on the models commonly used to study the pathophysiology of alcohol metabolism (please topic section 4). We added also the table (table 1) summarizing animal models with brief explanation in the revised manuscript.

Reviewer 2 Report

In the section 3.3 it would be useful toinsert more data on the transition between s tellate cells and myofibroblasts, the mechanisms of firosis formation and some possibilities of interfering with fibrosis formation.

Author Response

As you requested, we added more explanation for HSC activation and possible strategies for reducing liver fibrosis and HSC activation in the revised manuscript. 
